# Multi-, Inter-, and Transdisciplinarity within the Public Health Workforce: A Scoping Review to Assess Definitions and Applications of Concepts

**DOI:** 10.3390/ijerph191710902

**Published:** 2022-09-01

**Authors:** Kerstin Sell, Franziska Hommes, Florian Fischer, Laura Arnold

**Affiliations:** 1Institute of Medical Information Processing, Biometry, and Epidemiology, LMU Munich, Elisabeth-Winterhalter-Weg 6, 81377 Munich, Germany; 2Pettenkofer School of Public Health, 81377 Munich, Germany; 3German Network of Young Professionals in Public Health (NÖG), 80539 Munich, Germany; 4Institute of Public Health, Charité—Universitätsmedizin Berlin, Charitéplatz 1, 10117 Berlin, Germany; 5Bavarian Research Center of Digital Health and Social Care, Kempten University of Applied Sciences, Albert-Einstein-Straße 6, 87437 Kempten, Germany; 6Academy of Public Health Services, Kanzlerstraße 4, 40472 Duesseldorf, Germany; 7Department of International Health, Care and Public Health Research Institute—CAPHRI, Faculty of Health, Medicine and Life Sciences, Maastricht University, 6211 Maastricht, The Netherlands

**Keywords:** public health workforce, multidisciplinarity, interdisciplinarity, transdisciplinarity, collaboration, multiprofessional, public health, one health, global health

## Abstract

In light of the current public health challenges, calls for more inter- and transdisciplinarity in the public health workforce are increasing, particularly to respond to complex and intersecting health challenges, such as those presented by the climate crisis, emerging infectious diseases, or military conflict. Although widely used, it is unclear how the concepts of multi-, inter-, and transdisciplinarity are applied with respect to the public health workforce. We conducted a scoping review and qualitative content analysis to provide an overview of how the concepts of multi-, inter-, and transdisciplinarity are defined and applied in the academic literature about the public health workforce. Of the 1957 records identified, 324 articles were included in the review. Of those, 193, 176, and 53 mentioned the concepts of multi-, inter-, and transdisciplinarity, respectively. Overall, 44 articles provided a definition. Whilst definitions of multidisciplinarity were scarce, definitions of inter- and transdisciplinarity were more common and richer, highlighting the aim of the collaboration and the blurring and dissolution of disciplinary boundaries. A better understanding of the application of multi-, inter-, and transdisciplinarity is an important step to implementing these concepts in practice, including in institutional structures, academic curricula, and approaches in tackling public health challenges.

## 1. Introduction

Public health in the 21st century is faced with major global challenges and transformations, including persisting and increasing social inequalities and injustice [1,2,3], the climate and planetary health crises [4,5], war and conflict [6], as well as the burden of both communicable and non-communicable diseases (NCDs) [7].

These challenges are interlinked and share common features, in that they often have multiple causes and an inherent social complexity, are interdependent with other factors, and require multiple stakeholders to work on solutions; they thus share a set of characteristics which has been defined as making up a “wicked” problem [7,8]. In light of these intricacies, it is widely recognised that these current as well as future challenges cannot be solved by one discipline or profession alone but require multiple sectors, disciplines, and professions to work together by exchanging and combining their knowledge, expertise, and methods, as Rüegg et al. argue for the One Health approach [9]. It thus comes as no surprise that there has been an increase in calls for fostering and strengthening inter- and transdisciplinary work in issues related to public health.

The public health workforce, conceptualised as those professionals “primarily involved in protecting and promoting the health of whole or specific populations (as distinct from activities directed to the care of individuals)” [10], constitutes a heterogeneous workforce from diverse professional backgrounds that is tasked with public health service delivery and performing essential public health operations (EPHOs), as defined by the World Health Organization (WHO) [11]. The work context of the public health workforce is shaped by a set of particularities, including working at global, regional, national, local, and community levels on increasingly complex challenges such as those outlined above, cultural diversity, the evolution of diseases, working with public health stakeholders, private and public sector dynamics, inequities, financial crises, and an emphasis on addressing the social and economic determinants of health and infusing public health into the political agenda [12]. Internally, the public health workforce is faced with increasing personnel mobility and international collaboration [13], budgetary constraints, a professional identity crisis [7], an urgent need for more competency-based education and transformative learning [14], and better workforce retention [15,16], as well as personnel shortages, exacerbated during the COVID-19 pandemic [17]. For those in leadership functions in the public health workforce, interdisciplinary and interprofessional work along with collaborative, global, and digital work have been highlighted as fundamental capacities in the 21st century [12].

Hence, the relevance of multiple disciplinary work is strongly emphasised and evidently needed in public health, but references to these concepts often lack clarity, definition, and operationalisation. 

Multi-, inter-, and transdisciplinarity are terms often used to describe the specifics of multiple disciplinary teamwork, situated on a continuum of involvement [18]. Multiple disciplinary work in the context of (public) health is intended to address complex problems, to gather different perspectives, to develop testable hypotheses in research, to create definitions and guidelines, and to provide comprehensive health services and education (ibid). In particular, transdisciplinarity is seen as a method for researchers working on real-world problems, as prioritised by stakeholders and/or politicians [19]. 

Based on their extensive review of the academic literature and encyclopaedias, Choi and Pak [18] gathered definitions which state that **multidisciplinarity** “draws on knowledge from different disciplines but stays within the boundaries of those fields” [20], **interdisciplinarity** “analyses, synthesizes and harmonizes links between disciplines into a coordinated and coherent whole” [21], and **transdisciplinarity** “integrates the natural, social and health sciences in a humanities context, and in so doing transcends each of their traditional boundaries” [22]. For circumstances in which the degree of involvement of different disciplines is not clear, they suggest using the term “**multiple disciplinary**” [18]. 

Others have referred to the continuum of multiple disciplinary work as “cross-disciplinarity”, with uni-disciplinary work at one end and transdisciplinary work at the other end of the continuum [23].

Multi-professional work, in contrast, is a term often employed to describe the work of multiple professions working together within one discipline. The term is used in particular in a healthcare context, e.g., physiotherapists, physicians, and nurses collaborating, with knowledge from these different professions overlapping but not being integrated into each other’s practice [24,25,26]. 

In this article, the authors understand “disciplines” as overarching *fields of work*, for example, healthcare, law, or infectiology. “Professions” are understood as roles achieved through a specific training, academic education, or professional trajectory, e.g., a nurse, a solicitor, a hygiene inspector. 

Public health, understood as the “art and science of preventing disease, promoting health, and prolonging life through the organized efforts of society” [27] and as a discipline that thrives on the breadth of approaches [28], integrates multiple disciplines, such as epidemiology, environmental science, social work, political science, public administration, infectiology, and many more, depending on the scope of work [29,30]. Hence, public health constitutes an interesting case for examining multi-, inter-, and transdisciplinarity.

The rationale for this work is rooted in the pertinent observation that authors of the scientific literature about the public health workforce, both within and beyond a Special Issue guest edited by us [13], were often emphasising a need for interdisciplinary or transdisciplinary work to address public health challenges; however, a definition or description of these terms were rarely provided. As Jahn et al. observe: “Appeals for transdisciplinarity often do not spend much time explaining what they are precisely calling for—a fact that is also true of other forms of cross-disciplinary collaboration.” [31]. 

Hence, the objective of this scoping review was to investigate whether the three concepts of multi-, inter-, and transdisciplinarity are defined in academic literature about the public health workforce and to provide an overview of how the concepts are applied.

## 2. Materials and Methods

We conducted a scoping review, following the proposed framework by Arksey and O’Malley [32]. In addition to the scoping review steps, we conducted an abridged qualitative content analysis of a subset of articles to inform data charting. We report our results using the Reporting Items for Systematic reviews and Meta-Analyses extension for Scoping Reviews (PRISMA-ScR) Checklist [33].

### 2.1. Data Sources and Searches

We developed a search strategy around two themes, the public health workforce and the three concepts of multiple disciplinary work: 


*(“public health workforce”[Title/Abstract] OR “education public health professional”[Title/Abstract] OR “students public health”[Title/Abstract] OR (“public health”[Title/Abstract] AND (“profess*”[Title/Abstract] OR ”expert*”[Title/Abstract]))) AND (“multidisciplin*”[All Fields] OR “transdisciplin*”[All Fields] OR “interdisciplin*”[All Fields] OR ((“multiple”[All Fields] OR “multiples”[All Fields]) AND “disciplin*”[All Fields])) AND (english[Filter] OR german[Filter]).*


After piloting and refining the search strategy, we searched the scientific database PubMed on 6 May 2022. No further information sources were accessed. Records were retrieved from PubMed and imported in Rayyan, which is a web-based tool for conducting systematic reviews [34], and de-duplicated using Rayyan’s de-duplication function.

### 2.2. Eligibility Criteria 

Records were assessed for eligibility based on the following seven criteria:
**Population**: We included records referring to the public health workforce or public health professionals, defined as professionals who are “exclusively or substantially focused on issues related to population health in public health research, practice, policy, or education” ([13] adapted from [10]) and records referring to the collaboration of public health professionals with other professions. We excluded records focused on the health workforce, defined as professionals who are concerned with treating individual patients. Records about the collaboration of public health professionals with the general public were excluded as well.**Context—concept use**: Records that included at least one of the three concepts multi-, inter-, or transdisciplinarity in title, abstract, or keywords were included.**Context—concept application**: We included records that describe the application of at least one of the three concepts in reference to the public health workforce, e.g., on the following topics: ◦Extent and nature of multiple disciplinary work in public health workforce day-to-day practice (in reality or in an ideal scenario);◦Collaboration of different disciplines and professions in the public health workforce;◦Conditions, barriers, and facilitators for multiple disciplinary work;◦Necessity of multiple disciplinary work for public health work;◦Goals and aims of multiple disciplinary work;◦Training for multiple disciplinary work;◦Concepts, frameworks, or theories related to multiple disciplinary work;◦Areas of multiple disciplinary work. We excluded records that describe the application of at least one of the three concepts without focusing on relevant aspects for the public health workforce, for example:◦Articles about multiple disciplinary work to address one specific health problem in a clinical context (e.g., treatment of fatty liver disease or pain), as opposed to addressing a broader public health area;◦Articles predominantly focused on healthcare;◦One-off consultation of public health professionals, e.g., for planning rehabilitation services or as interview participants in studies.
**Language**: Records in English and German were included.**Publication type**: Academic literature was included. Hence, we included empirical articles, (systematic) reviews, and opinion pieces. Guidelines were included when they were published in peer-reviewed journals. Editorials, animal studies, conference abstracts, and book chapters were excluded. Clinical trials were excluded as they were unlikely to address a broad public health topic.**Publication date**: There were no restrictions based on publication date.**Full-text not accessible**: We excluded records for which we were not able to obtain the full text.


### 2.3. Screening

All authors screened the titles and abstracts of the same 50 articles and discussed discrepancies to calibrate the screening process and clarify inclusion and exclusion criteria. After the calibration phase, all remaining records were screened by one reviewer and articles labelled as “unclear” were screened in duplicate and discussed among two authors until consensus on eligibility was achieved. All records that were included at this stage were double screened in order to confirm inclusion and relevance for the review question. A combined full-text screening and data extraction were undertaken. At this stage, articles that did not meet the inclusion criteria were excluded. Articles for which we were not able to obtain the full text through our institutional accounts were excluded as well.

### 2.4. Data Charting of First-Level Articles

A data extraction sheet was developed and subsequently piloted with every author undertaking data extraction of at least twenty different articles in order to test usability, comprehension, and relevance of data extraction categories. This process simultaneously served as a calibration exercise for the data extraction stage. After team discussion and further adaptation, the data extraction form included the following categories: basic information about the record (title, author, year of publication, and link to online resource), specification of the concepts mentioned in the article (multi-, inter-, and/or transdisciplinarity), provision of a definition for the concept(s) (no; yes, citation; yes, own definition; yes, both citation and own definition), further information about the definitions (extracted quote; author, year of reference cited; link to cited reference), and further article information and details about the content of the article (country of the first author, study type, main disciplinary focus of the article (e.g., public health, one health, global health, etc.), broader public health topic, main results, involved professions, and whether the concepts were applied within or beyond the public health workforce). Included records were organised into batches of similar sizes and divided up among the review authors; full-text screening and data extraction were then conducted by one review author with a second author undertaking consistency checks. 

We extracted data for all data extraction categories for those articles that did provide a definition for one (or more) of the three concepts, multi-, inter-, and transdisciplinarity. For the articles that did not provide a definition, we only extracted basic information. In order to calibrate our coding, we discussed all ambiguous passages among two reviewers and conducted thorough consistency checks.

In order to streamline the full-text screening and data extraction process, we searched articles for the three concepts using the document search function for “discip” to ascertain whether or not a definition was provided. We considered a definition provided (a) when the authors described the ways of working together of different disciplines in public health, (b) when a citation was provided after one of the three concepts was mentioned, or (c) when both a description and a citation were available. A list of professions described as working together was not considered a sufficient definition as it lacked detail about the how of working together [35]. We then checked the cited references to ascertain whether they provided a definition of one of the three concepts, which we understood, again, as a description of the ways of working together among multiple disciplines. If the cited reference only provided another citation to define the concept(s), this did not meet our criterion for “definition provided” of the primary article. Hence, we considered a definition to be available only if the authors spelled out a definition and/or if they provided a citation that included a written definition of the concept(s). Cited articles that meet these inclusion criteria form the second level of analysis (Figure 1).

### 2.5. Data Charting and Analysis of Second-Level Articles

After data extraction of eligible articles identified in our systematic database search (“first level articles”), a second data charting exercise was undertaken for articles that we identified as references providing definitions of multi-, inter-, and transdisciplinarity, that were cited to define the concepts (“second level articles”). We developed a second data extraction sheet in order to extract basic information about these articles, which were not restricted by any further exclusion criteria such as those defined for the first-level articles. We double-checked whether these second-level articles did spell out a definition of multi-, inter-, and/or transdisciplinarity. For articles that did not provide a definition meeting our criteria, we excluded the respective article from the second-level data charting and relabelled the first-level article, citing this reference as providing “no definition”. For example, a WHO report, cited by Ghanbari et al. [36], was excluded from the second-level analysis as it defined multidisciplinary too simplistically as “multiple disciplines are working together (i.e., in a single ministry or research institute employing physicians, nurses, veterinarians, epidemiologists, laboratory scientists, basic scientists, and/or other health professions)” [37]. In another article, interview excerpts included allusions to definitions of the three concepts, which we excluded as the definitions were not provided by the study’s authors themselves [38].

Subsequently, we read the second-level articles and reports thoroughly to identify overarching concept components making up the definitions of multi-, inter-, and transdisciplinarity, drawing on the theoretical literature represented by those records. The concept components were iteratively derived from identified categories that encompassed the various attributes and characteristics of the definitions of multi-, inter-, or transdisciplinarity. Following an inductive approach, one author first extracted relevant text passages related to definitions of one of the three concepts and iteratively assigned them to emergent attributes and features, which were cross-checked by a second author and ultimately consolidated into different concept components in a second step. The list of concept components was then reviewed and discussed by all review authors, subsequently condensing the list. For example, we identified and defined the concept component “disciplinary boundaries” as a distinct category of characteristics, such as the blurring or maintenance of boundaries that were relevant for defining multi-, inter-, and transdisciplinarity. Due to time constraints, we did not consult the books listed in our overview of second-level articles.

### 2.6. Data Analysis of First-Level Articles

Following identification of the concept components of definitions of multi-, inter-, and transdisciplinarity, we analysed the extent to which these components were present in the previously extracted definitions of multi-, inter-, and transdisciplinarity from the first-level articles. First, we coded whether a given concept component was addressed in the definition extracts. Subsequently, we analysed the extracts for each concept component for similarities, recurring themes, and aspects with particular relevance to the public health workforce. This information was tabulated and summarised narratively.

Because no sensitive data were analysed, data extraction was undertaken in Google Spreadsheet, which allowed for simultaneous collaborative work. We provide descriptive statistics of the charted data and a narrative illustration of the results of the qualitative analysis.

## 3. Results

Our systematic search rendered 1957 records, of which 1947 were subsequently screened for eligibility after de-duplication, resulting in 371 articles that were included for the full-text screening. Ultimately, 324 articles were included in the review (Figure 2).

### 3.1. Data Charting of First-Level Records

Overall, the concepts of multi-, inter-, and transdisciplinarity were mentioned widely in the 324 articles included in the review: Multidisciplinarity was referred to in 193 articles, interdisciplinarity in 176, and transdisciplinarity in 53 articles. Table 1 gives an overview of total, single, and double mentions of the concepts. Multi- and interdisciplinarity were often referred to as single concepts and in combination. There were more articles that referred to both inter- and multidisciplinarity (*n* = 59) than articles that referred to either other constellation of the concepts (*n* = 11 and *n* = 12, respectively). Fourteen articles mentioned all three concepts.

Of the 324 articles included in this review, 280 did not provide a definition of multi-, inter-, or transdisciplinarity that met our criteria. Of the 44 articles providing a definition, the authors of 18 articles cited a reference that included a definition, and in 4 articles, authors stated their own definition, without providing a reference. In the remaining 22 articles, authors spelled out a definition of one or more of the concepts and cited a reference providing a definition (Table 2). 

Of those articles mentioning a single concept, the definitions were provided in 6 out of 109 (6%) [30,36,37,38,39,40,41,42,43], 10 out of 91 (11%) [28,44,45,46,47,48,49,50,51,52], and 9 articles out of 16 (56%) [53,54,55,56,57,58,59,60,61] mentioning multi-, inter-, and transdisciplinarity, respectively (Table 2, Figure 3). 

Of the 59 articles referring to both multi- and interdisciplinarity, only 8 (14 %) provided a definition for one or both concepts [26,62,63,64,65,66,67,68]. Of the twelve articles referring to inter- and transdisciplinarity, four provided a definition for the concept(s) [9,69,70,71], and among the eleven articles mentioning multi- and transdisciplinarity, only one provided a definition [72].

A total of fourteen articles mentioned all three concepts [38,73,74,75,76,77,78,79,80,81,82,83,84,85,86,87] and six of these articles provided a definition for one or more of the concepts [73,74,76,77,82,84].

**Table 2 ijerph-19-10902-t002:** Overview of first-level articles with definitions and linked second-level articles.

First-Level Articles					Cited References (Second-Level Articles)				
First Author Year	Definition Provided	MD	ID	TD	First Author Year	Article Type	MD	ID	TD
Aguirre 2016 [69]	own and citation	no	yes	yes	Aguirre 2008 [88]	Editorial	no	no	yes
Amuguni 2017 [73]	citation	yes	yes	yes	Conrad 2013 [87]	Conceptual paper	yes	yes	yes
Bartonova 2012 [70]	own and citation	no	yes	yes	Aboelela 2007 [35]	Original article	yes	yes	yes
					Guimarães 2006 [89]	Conceptual paper	no	no	yes
					Pohl (for Network for Transdisciplinary Research) [90]	Website	no	no	yes
Bay 2017 [62]	citation	yes	yes	no	Perkins 2014 [91]	Book			
Begg 2015 [48]	citation	no	yes	no	Von Hartesveldt 2008 [92]	Report	no	yes	no
Bolduc 2007 [47]	citation	no	yes	no	Klein 1990 [93]	Book			
Broyles 2013 [74]	citation	yes	yes	yes	Madden 2006 [94]	Original article	no	yes	no
Chan 2019 [46]	citation	no	yes	no	Choi 2006 [18]	Review	yes	yes	yes
Choi 2021 [63]	citation	yes	yes	no	Choi 2006 [18]	Review	yes	yes	yes
D’Alessandro 2017 [76]	citation	yes	yes	yes	Lawrence 2004 [95]	Conceptual paper	yes	yes	yes
Ding 2022 [77]	citation	yes	yes	yes	Davé 2016 [96]	Report	no	yes	no
					Stipelman 2014 [97]	Letter	yes	yes	yes
El Ansari 2003 [30]	citation	yes	no	no	Naidoo 2001 [98]	Book chapter			
Evans 2003 [42]	citation	yes	no	no	Baggott 2000 [99]	Book			
					Beaglehole 1997 [100]	Book			
					Cowley (Ed.) 2002 [101]	Book			
					Griffiths (Ed.) 1999 [102]	Book			
					Levenson 1999 [103]	Book chapter			
					McPherson 1997 [86]	Book chapter			
Fam 2017 [55]	own and citation	no	no	yes	Jahn 2012 [31]	Conceptual paper	no	no	yes
					Pohl 2010 [104]	Conceptual paper	no	no	yes
Ferreira-Neto 2016 [26]	own and citation	yes	yes	no	Leite 2013 [105]	Original article			
Fischer 2005 [72]	own definition	yes	no	yes	NA				
Forbat 2015 [49]	own and citation	no	yes	no	Repko 2008 [106]	Book			
Ghanbari 2020 [36]	citation	yes	no	no	Conrad 2013 [87]	Conceptual paper	yes	yes	yes
Gosselin 2011 [56]	own and citation	no	no	yes	Lee 2007 [107]	Book chapter			
					Nicolescu 2008 [108]	Book			
					Stokols 2006 [109]	Conceptual paper	yes	yes	yes
Greacen 2012 [50]	own and citation	no	yes	no	unknown	Unclear			
Holst 2021 [64]	own and citation	yes	yes	no	Choi 2006 [18]	Review	yes	yes	yes
					Fiore 2007 [110]	Book chapter			
					National Academy of Sciences, National Academy of Engineering, Institute of Medicine 2005 [111]	Book			
Horowitz 2017 [57]	own and citation	no	no	yes	Leshner (Ed.) 2013 [112]	Book			
					Nash 2008 [113]	Conceptual paper	yes	yes	yes
James 2015 [58]	own and citation	no	no	yes	Hiatt 2008 [114]	Conceptual paper	yes	yes	yes
Kavanagh 2015 [51]	own and citation	no	yes	no	MCH Leadership Competencies Workgroup (Ed.) 2009 [115]	Report	no	yes	no
Kent 2012 [45]	citation	no	yes	no	Kent 2011 [116]	Report	no	yes	no
Lapaige 2010 [59]	own and citation	no	no	yes	Hirsch Hadorn (Ed.) 2008 [117]	Book			
Lief 1968 [65]	own definition	yes	yes	no	NA				
Lloret 2020 [60]	own and citation	no	no	yes	Thompson 2017 [118]	Original article	no	no	yes
Lucchini 2018 [66]	own and citation	yes	yes	no	Erickson 2016 [119]	Conceptual paper	yes	yes	no
Manyara 2018 [43]	own and citation	yes	no	no	Choi 2006 [18]	Review	yes	yes	yes
Marcotty 2013 [71]	own and citation	no	yes	yes	van Manen 2001 [120]	Comment	no	yes	yes
Margolis 2012 [52]	own definition	no	yes	no	NA				
Marshall 2011 [54]	citation	no	no	yes	Syme 2008 [121]	Comment	no	yes	yes
Nash 2003 [82]	own and citation	yes	yes	yes	Morgan 2003 [122]	Conceptual paper	no	no	yes
					Pellmar (Ed.) 2000 [123]	Book			
					Rosenfield 1992 [124]	Conceptual paper	yes	yes	yes
					Stokols 1998 [125]	Unclear			
Orme 2007 [41]	citation	yes	no	no	McPherson 1997 [86]	Book chapter			
					Bell 2003 [126]	Book chapter			
					Evans 2005 [127]	Report			
					Evans 2003 [42]	Conceptual paper	yes	no	no
Ramanathan 2017 [53]	citation	no	no	yes	Burris 2017 [128]	Conceptual paper	no	no	yes
Razum 2015 [28]	own definition	no	yes	no	NA				
Rüegg 2017 [9]	own and citation	no	yes	yes	Choi 2006 [18]	Review	yes	yes	yes
					Schelling 2015 [129]	Book chapter			
					Zinsstag 2012 [130]	Editorial	no	no	yes
Taub 2003 [84]	own and citation	yes	yes	yes	McCallin 2001 [131]	Review	yes	yes	no
Umble 2003 [44]	citation	no	yes	no	Sorenson 1991 [132]	Book			
Vamos 2012 [61]	own and citation	no	no	yes	Rosenfield 1992 [124]	Conceptual paper	yes	yes	yes
White 2013 [67]	own and citation	yes	yes	no	Choi 2006 [18]	Review	yes	yes	yes
Williamson 2004 [40]	citation	yes	no	no	DoH 1999 [133]	Policy paper	yes	no	no
Yamada 2007 [68]	own and citation	yes	yes	no	Advisory Committee onInterdisciplinary, Community-Based Linkages 2001 [134]	Report	yes	yes	no

Abbreviations: MD = multidisciplinarity, ID = interdisciplinarity, TD = transdisciplinarity. The table gives an overview of concepts mentioned in the first- and second-level articles. Books and book chapters included in the list of second-level records were not examined regarding the definitions of multiple disciplinary work due to accessibility and time constraints. Therefore, there is no information regarding the concepts mentioned in those references available.

The broad majority of the articles mentioning multi-, inter-, and/or transdisciplinarity were published after 2000 and we found temporal differences regarding the publication date of articles mentioning the three concepts: On average, articles mentioning multi-, inter-, or transdisciplinarity were published in 2012, 2013, and 2015, respectively. Of all the articles that could be analysed according to our inclusion criteria, only one article was published before 2000 [65]. In the article, both multi- and transdisciplinarity were mentioned and the authors gave a definition for at least one of the concepts [65]. 

The majority of the articles providing a definition of multiple disciplinary work were from first authors affiliated to an institution in the United States of America (*n* = 17), followed by six articles from authors in the UK, five from Canada, and ten articles from eight other European countries (more detailed information can be found in the Appendix A). Of those, two articles were from Germany-affiliated authors, one of which was in German. All the articles by UK-affiliated authors mentioned multidisciplinarity and four of these articles mentioned multidisciplinarity exclusively, whilst authors from the other regions mentioned all three concepts or different variations.

### 3.2. Data Charting and Analysis of Second-Level Records

Linked to the 40 articles providing one or more citations (either as standalone citations or linked to a description of the concepts) to define multi-, inter-, and/or transdisciplinarity, we identified a total of 55 distinct references/second-level articles and other records. Of those, there were 21 books or book chapters, 15 conceptual papers, 6 reports, 4 original articles, 2 reviews, 2 editorials, and 1 website, policy paper, and letter each. Furthermore, we identified two references with unclear document types for which we were not able to obtain the full text. 

One conceptual paper [87] was cited twice, and one review [18] was cited six times in articles included in our review. We were able to verify for all the references except for the books and for those for which we were not able to obtain the full text that the reference did indeed provide a spelled-out definition of multi-, inter-, and/or transdisciplinarity. After excluding the 21 books, book chapters, foreign-language records, and those without full-text access, we were able to analyse a total of 31 individual second-level articles and reports. 

Based on the definitions of multi-, inter-, and transdisciplinarity provided in the second-level articles, we identified five concept components that were often mentioned when describing and/or defining the three concepts in this rich theoretical literature. Drawing in particular on the work of Rosenfield [124] and Choi and Pak [18], these concept components include the *involved disciplines and/or professions*, *mode of collaboration*, *aim and purpose*, *role of participants*, and *disciplinary boundaries*. A short definition of each concept component is provided in Table 3.

### 3.3. Analysis of First-Level Articles and Definitions

In our analysis of the definitions extracted from the articles included in our review (*n* = 26, four articles with “own definitions” and 22 definitions drawing on other literature), we found that the concept components *aim and purpose* as well as *mode of collaboration* were most commonly used in the definitions of multi-, inter-, and transdisciplinarity. 

#### 3.3.1. Involved Disciplines and/or Professions

In the extracted definitions in the literature about the public health workforce, the disciplines and professions involved were predominantly described in terms of the diversity of the participants involved, with authors often stating that individuals from “multiple” or “two or more disciplines” worked together in multi-, inter-, and transdisciplinary work [52,60,61,64,70,84]. The list of disciplines was more detailed for inter- and transdisciplinarity, highlighting the involvement of scientists with practitioners [69], non-academics [60], and societal groups [9] in transdisciplinary work and the collaboration of scientists with community members and “consumers” in interdisciplinary work [51,52]. Here, Bartonova [70] also explicitly includes policy-makers as stakeholders to be addressed, as strong political support is considered a prerequisite for addressing complex public health issues. For multi- and interdisciplinary work, “professionals” were also listed as participants, whilst there was a stronger emphasis on scientists collaborating with other groups for inter- and transdisciplinary work. For the extracted definitions of transdisciplinarity, there was a further emphasis on groups of individuals from different epistemological positions collaborating, as evident in the mentioning of “decision-makers and knowledge users” [56], diverse “knowledge bases” [60], and the “integration of society and science by including all stakeholders” [9].

#### 3.3.2. Mode of Collaboration

With respect to the concept component mode of collaboration, the extracted definitions of multidisciplinarity were focused on the different disciplines or professions “contributing” knowledge and skills [64,67,135], working towards a common goal, orientation, or problem [64,65,72], and an “alignment” of professions [65].

In definitions of interdisciplinarity, the collaboration of disciplines was described in terms of co-responsibility ([26] citing [105]), collective ownership [50] as well as being integrative, synergistic, and synthesising or harmonising efforts [51,64,67] to achieve a “coherent whole” [67] or “coherent entity” [64]. Other definitions remained vague, describing the work mode merely as collaboration [52,68,70].

In the definitions of transdisciplinarity, the integration of the skills, perspectives, and expertise of different disciplines [9,58,60,82,84] and the coherent whole emerging from these efforts as an outcome also featured prominently. In addition to the characteristics mentioned in the definitions of interdisciplinarity, however, authors also emphasised the “participatory” nature of transdisciplinary work [57,60,84] and its conceptualisation as knowledge co-production or knowledge translation [59,60].

#### 3.3.3. Aim and Purpose

The majority of the extracted definitions of inter- and transdisciplinarity referred to the aim and purpose [9,28,49,50,51,52,55,56,57,58,59,60,61,64,66,68,69,70,71,82,84], while only one extracted definition mentioned a purpose of multidisciplinary work [72]. 

A major aim described for transdisciplinarity was to integrate knowledge, expertise, as well as methods from the different team members [58,69,104] and to set goals in a participatory manner [57]. This integrative and holistic approach was described as enabling the creation of more efficient and comprehensive interventions [69,71] and “allows for an integrated view with completely different outcomes from what one would expect from just the addition of the parts” ([60] citing [136]).

The aim of transdisciplinary work was also specified by its innovative character, enabling new discoveries ([57] citing [137]) as well as “conceptual frameworks and methodological tools” ([82] citing [124]). 

Problem solving “for what is perceived to be the common good” [59] was mentioned as another major aim of transdisciplinarity in public health. The respective problems were characterised as “socially relevant” [104], as “complex, ill-defined problems concerning human–environment interactions” [56], or “problems of the lifeworld” [59]. The major public health challenges described by authors as requiring a transdisciplinary approach were climate change [9,56], infectious disease control [71], drug resistance, food and water security and safety [9], as well as cancer [58].

Similarly, the definitions of interdisciplinarity referred to the aim of solving “community or system problems” [138] or as problems for which “solutions are beyond the scope of a single discipline or area of research” ([64] citing [111]). These were further characterised as “prevention of environmentally related diseases” [70], safety performance [66], mental health promotion [50], and obesity reduction [52]. 

Another purpose of interdisciplinarity mentioned by Razum [28] was to contribute to public health by being close to reality and relevant for policy and practice.

In a definition of multidisciplinarity, it was stated that the involved team members aim to address the same problems [72] without further specifying the character of these problems.

#### 3.3.4. Role of Participants

Only a few extracted definitions referred to the role of participants in multidisciplinary endeavours. However, these definitions emphasise that despite a collaboration among different disciplines, scientists need to “maintain their levels of specialization” [72] by contributing knowledge and skills from their respective discipline [67]. The research activities may either focus upon questions of interest to individual investigators or are shared among different disciplines [84].

On the contrary, the definitions of interdisciplinarity highlighted that professionals from different disciplines are aware of what the specificity of their discipline is [26] but also accept co-responsibility across disciplines ([26] citing [105]) for answering questions which are important to the involved parties [84]. In doing so, interdisciplinary scientists are free to communicate with and coordinate the work of specialised scientists from disparate fields ([82] citing [139]) but may not integrate approaches from these fields in their own work [82]. The respective skills and expertise of team members from different disciplines [51], including shared leadership skills [50], are described as essential and synergistic [51]. The valuing of the expertise and skills of all team members [51] and equality [28] are further highlighted in the definitions of interdisciplinarity.

The definitions of transdisciplinarity point to the requirement to be “educated and attuned to the nuances and best practices for working in teams” [57]. This includes learning the languages, cultures, and norms of other disciplines [58]. In transdisciplinarity, overcoming “disciplinary silos” [56] may be achieved by shared roles [69]. Here, the differentiation of the roles is not defined by discipline-specific characteristics but by the needs of the situation [69]. According to this, the theories and methods of the involved disciplines are to be adopted to the needs of other disciplinary team members. This indicates that each team member needs to be able to transcend their individual perspective [84]. Thereby, new conceptual frameworks and methodological tools arise as an integration of individual disciplinary perspectives ([140] citing [124]).

Across the definitions of all three concepts, it is emphasised that the multiple disciplinary work requires time and communication skills [57,69,72].

#### 3.3.5. Disciplinary Boundaries

Finally, within the conceptual component of disciplinary boundaries, we extracted the descriptions and approaches that address the maintenance, blurring, or dissolution of individual professional perspectives or methods of multidisciplinary work.

Overcoming often rigid discipline-specific definitions of phenomena and questioning them on the basis of the know-how and methodological approach of other disciplines was frequently mentioned in the analysed literature as central to solving complex public health challenges and real-world problems [28,56,66,84]. 

Within the included public health workforce literature, disciplinary boundaries were predominantly discussed in the context of transdisciplinary approaches, partly also in connection with interdisciplinary approaches. 

Within multidisciplinary approaches, collaboration across professional boundaries mainly referred to the fact that professionals with different backgrounds bring their (specific) knowledge and disciplinary skills to one profession [43].

Interdisciplinary approaches were conceptualised to go a step further, and the explicit need to develop collaborative approaches to problem solving by going beyond one’s one discipline was emphasised [68]. In this context, interdisciplinarity was defined as crossing boundaries [84]. The integration of knowledge from different disciplines [66] is hereby considered just as crucial for goal-oriented and equal collaboration as the continuous clarification of the meaning of the terms used and agreement on a common theoretical framework [28]. At the same time, this requires a strong commitment from all stakeholders, as the complexity of each discipline needs to be understood and respected by all.

Finally, transdisciplinary approaches are described as fully dissolving traditional boundaries by crossing disciplinary paradigms [55,59] and integrating knowledge and perspectives from scientific and non-scientific sources alike to develop more holistic approaches that bridge ecosystem and human health boundaries [69,72]. James [58] sees the great advantage of transdisciplinary approaches in the fact that by breaking down rigid discipline-specific definitions of phenomena and critically questioning them, more comprehensive definitions, understandings, and connections can be developed that are more appropriate to the complex reality.

#### 3.3.6. Other

Beyond these five common concept components, the authors also commented on the requirements, in particular for transdisciplinary work: Time and energy [56] are a prerequisite, as well as the need to grasp complex issues and dimensions [59] in a “working environment [which] is intellectually rich and challenging” [67].

## 4. Discussion

### 4.1. Summary of Findings

We identified 324 articles mentioning multi-, inter-, and/or transdisciplinarity in our scoping review of articles about the public health workforce. Only 44 of these articles provided a definition of one or more of these concepts, either through citation and/or by providing a description of the concepts. Multi- and interdisciplinarity were *mentioned* most commonly, whilst transdisciplinarity was more likely to be *defined* in the included articles. Our analysis of the concept components of the definitions available in these articles demonstrated that the concepts were often described in terms of the involved disciplines, the mode of collaboration, and the aim and purpose of the multiple disciplinary work. Whilst multidisciplinarity was somewhat under-defined in the included articles, the definitions of inter- and transdisciplinarity were richer, emphasising in particular the blurring or dissolution of disciplinary boundaries and the goals of such endeavours in public health.

### 4.2. Quantitative Results

As we hypothesised, a majority of the articles about the public health workforce that referred to multi-, inter-, and transdisciplinarity did not define or describe those concepts. This may be indicative of a lack of conceptual clarity among authors and could be related to the variety of definitions available, as others have noted [31,96]. This assumption is further supported by our finding that in particular multi- and interdisciplinarity were frequently mentioned together but rarely defined, indicating an interchangeable, conceptually non-distinct use.

In some cases, the concepts seemed to be used as catch phrases without further reference to the concrete operationalisation of multiple disciplinary work, such as resource implications and practicalities. Fuest [141] has observed this previously, for example, in relation to interdisciplinary research.

Among the few articles that provided a definition, transdisciplinarity is interestingly more commonly defined than multi- and interdisciplinarity. This may be due to its perceived “novelty”, also indicated by its more recent emergence as our analysis shows, particularly in the field of One Health [69]. Potentially, its boundary-spanning nature and higher conceptual complexity may also prompt authors to provide a definition, citation, or otherwise clarify the concept. Another reason for the lower number of articles providing definitions for multidisciplinarity may be due to our inclusion criteria: we excluded articles that only provided a list of individuals involved in multiple disciplinary work but no further definition details. In particular, multidisciplinarity appears to have often only been described in terms of the involved parties, which led to exclusion. 

A different explanation for the lack of definitions of the three concepts may be quite the opposite: Authors may be very familiar with the concept or consider themselves very familiar with the concept and assume that their readership will also be familiar with the terms. 

In this context, we do find it important to emphasise that not all academic literature needs to define the terms multi-, inter-, and transdisciplinarity, and this article is not meant to be understood as normative, valuing articles that provide a definition for the concept(s) over articles without definitions. In particular, multidisciplinarity may be easily and intuitively the default mode of collaboration readers may have in mind. However, in-depth discussions about the ways of working together in the public health workforce may benefit from more conceptual clarity and precision.

### 4.3. Qualitative Results

In our qualitative analysis of the concept component “**involved disciplines and/or professions**”, scientists were often referred to as key players in inter- and transdisciplinary work, an observation which was also pertinent in the more theoretical, second-level articles [31,124]. This is likely due to the fact that we only consulted academic literature that was listed in PubMed and the inclination of scientists to write about their own work. Public health practice, however, depending on its organisational level, does not necessarily always involve scientists. The public health service in Germany (PHS, in German: Öffentlicher Gesundheitsdienst), for example, is made up of professionals from diverse backgrounds and disciplines, including professionals with foundational training as well as professionals with graduate education. In Germany and elsewhere, this heterogeneous group is undertaking multiple disciplinary work within and beyond the PHS, with professionals from social work, veterinary medicine, biology, urban planning, among others [142,143,144]. Thus, it appears that there is an imbalance between the wealth of academic literature about multiple disciplinary work involving public health scientists and a scarcity of academic literature investigating multiple disciplinary work in public health practice. 

Another group mentioned included decision-makers and “knowledge users” in transdisciplinary work [56] and policy-makers in interdisciplinary work [70], demonstrating links with knowledge to action concepts and frameworks [145,146], which we will discuss later. 

The “**mode of collaboration**” in the definitions of multiple disciplinary work was described as a continuum ranging from individuals from different disciplines “contributing” expertise [43,64,67] in multidisciplinary approaches to integration, synthesis, and building a “coherent whole” in inter- and transdisciplinary approaches. The latter were often referred to in similar terms, indicating a substantial conceptual overlap. Transdisciplinarity has been described as an extension of interdisciplinary forms of the problem-specific integration of knowledge and methods [31], while multidisciplinarity appeared as a more distinct and decisively less synergistic and integrative mode of collaboration. 

With respect to the public health workforce, the work on the EPHOs may lend itself to all three types of multiple disciplinary work, each with its respective value and usefulness for different essential public health operations. For example, many public health surveillance functions (EPHO 1) require multidisciplinary work, combining the expertise and work of clinicians and laboratories identifying diseases with that of epidemiologists analysing the resulting datasets. Advocacy, as part of EPHO 9, may require the interdisciplinary work of public health specialists integrating their expertise with that of communications experts and political scientists to achieve policy impact. As White [67] observed, “Public health is multidisciplinary because the many types of professionals in this field contribute knowledge and skills from their own disciplines, yet it is also interdisciplinary because its practitioners must analyze, synthesize and harmonize links across disciplines into a coherent whole”. 

Authors writing about the public health workforce also noted the participatory nature of transdisciplinary work [55,57,60], which we will discuss below as a phenomenon linked to knowledge co-production approaches. 

The integration and synthesis of knowledge, expertise, and skills was not only described as a mode of collaboration but also as an “**aim and a purpose**” of inter- and transdisciplinary work. Across a vast majority of the articles, the aim of these approaches is described as addressing complex problems through the concepts’ innovative potential for creating new ideas, tools, methodologies, processes, and solutions [57,82,124,137]. The complexity of problems public health professionals face have led to their labelling as “wicked” problems [7,8] that require multiple disciplines to develop solutions. 

These concepts do not constitute new concepts to addressing complex problems: Efforts to strengthen interdisciplinary research in, e.g., social psychology and agriculture, have been initiated as early as post-World War II, but sustainability beyond singular one-off-projects has often been poor, as Rosenfield [124] argues. Transdisciplinary research in sustainability science undertaken as early as the 1960s and 1970s was equally difficult to sustain [118]. In the health field, the three concepts were first mentioned together in 1983 and used interchangeably by authors, as Choi and Pak [18] found in their MEDLINE search. Whilst many disciplines have experienced an increasing specialisation and narrowing of the disciplinary scope in the past decades, it has been argued that public health has not become a hyper-specialised discipline but rather a pluralist umbrella discipline, implying its inherent interdisciplinarity [28]. 

In their substantive work on analysing the state of health and public health professional education, Frenk et al. [14] argue that public health education needs urgent reform in order to prepare the discipline for agile and rapid adaptation and to respond to the speed of global change and transformations [14], without mentioning inter- and transdisciplinary work as a field for which education and training are required. This contrasts with the large number of articles we identified that framed inter- and transdisciplinary work as key solutions to tackling global transformations and challenges.

Accordingly, the “**roles of participants**”, including the associated responsibilities and competencies of those involved, have been described differently in the three concepts of multi-, inter-, and transdisciplinarity: While in multidisciplinary undertakings the individuals involved contribute their expertise and skills primarily while maintaining their respective specialisations [72], inter- and transdisciplinary approaches deliberately break down these silos to address multiple issues [26,82,84]. Many authors noted that successful inter- and transdisciplinary work requires specific skills that enable both researchers and practitioners to go beyond their individual perspectives [84]. In addition to extensive communication skills and a willingness to engage with new and different approaches, this also requires flexibility, openness, and a low-hierarchy, appreciative way of working together [12,147,148].

The respective leadership level has a special role to play here: In order to address the key health challenges at all levels—global, national, and local—“horizontal approaches” are needed that create alliance-based relationships for collaboration between groups of individuals instead of the top-down leadership approaches that have long prevailed [12]. From this, a call to action can also be derived for public health schools and other institutions of public health workforce education to consider how the skills and competencies needed for this can already be acquired in (higher) education. Given the strong practical relevance of complex public health interventions, it is also conceivable that the skills needed to acquire leadership competencies could be firmly integrated into practice-oriented training concepts. Finally, both could have a positive effect on the recruitment of new professionals: White [67] emphasises that it is precisely the multi- and transdisciplinary elements in the context of public health that make the work attractive and interesting for (young) professionals—training and education elements included.

The maintenance, blurring, and transcendence of “**disciplinary boundaries**” constitute another component of the definitions of multi-, inter-, and transdisciplinarity. With respect to this concept component, transdisciplinarity was most commonly defined in the articles about the public health workforce and described as a dissolution of traditional boundaries [55,59] and the integration of knowledge from scientific and non-scientific sources [69,72]. 

This aspect of transdisciplinary work shares characteristics with knowledge co-creation [118], research co-production, engaged scholarship, and integrated knowledge translation (IKT), a set of approaches intended to produce research evidence and other knowledge with citizens, stakeholder groups, decision-makers, and other knowledge users through a participatory process [149,150]. Beyond the transcendence of epistemic disciplinary boundaries, these efforts are considered more democratic approaches to science [118] and as elements of science governance, increasing scientific accountability to the public [19]. IKT, an active, ongoing collaboration between researchers and decision-makers, is intended to increase access to scientific evidence in decision making and enhance the policy relevance of research [150]. In addition to these benefits, Burris et al. [128] hypothesise practitioners’ improved access to insights from implementation research as practical benefits of transdisciplinarity for the particular case of public health law and legal epidemiology, arising from links between research, advocacy, and practice. Transdisciplinarity and other co-creation approaches hence share a set of assumed benefits, along with a shared set of challenges.

Some of these challenges were touched on within definitions of multiple disciplinary work and subsumed in our sixth category, in particular related to the resources required to undertake this work, e.g., time and energy [56]. In addition to the challenges related to insufficient time or funding, Choi and Pak [147] further identified a lack of leadership and poor team composition, institutional constraints, discipline conflicts, and unequal power between disciplines as hurdles to multiple disciplinary work. Their work highlights the opportunity cost of the multiple disciplinary cost and the need to carefully weigh advantages and disadvantages when planning such endeavours. Hence, it is valuable to carefully choose an appropriate multiple disciplinary work mode, as suggested by Whitfield and Reid [151], and making it explicit when planning public health projects, organisational structures, and managing teams.

### 4.4. Reflexivity

All four authors are members of the German Network of Young Professionals in Public Health (Nachwuchsnetzwerk Öffentliche Gesundheit, NÖG, https://noeg.org/ (accessed on 31 July 2022)), an informal network of people with an interest in public health from the perspective of young professionals [39]. Amongst other issues, the network advocates for strengthening interdisciplinary work and the role of professionals from diverse backgrounds in the public health service in Germany. Hence, this work has shaped our view of multiple disciplinary work and sparked in particular our interest in studying these concepts more in-depth, as we find it ourselves challenging to define the concept of interdisciplinarity in our own practice and publications. Whilst we believe that there is great value in interdisciplinary work, we do not consider one of the concepts more valuable than the others but see the three concepts as complementary and acknowledge their respective value and usefulness for different tasks and aims.

### 4.5. Strengths and Limitations

Our work’s main limitation lies in its exclusive focus on the three concepts, multi-, inter-, and transdisciplinarity. We did not explore similar or overlapping concepts, such as interprofessional education [152], interprofessional practice [153], multi-professional collaboration, intersectoral work, pluridisciplinarity, and cross-disciplinarity [154]. We cannot rule out that our overall findings and conclusions would have turned out slightly different when taking all these concepts into account. However, based on our experience in working in and with the public health workforce, the three investigated concepts can be considered as major ones. 

The scope of this project also did not allow us to examine the definitions of multiple disciplinary work provided in books, such as the work of Julie Klein [93], which would have likely enriched our work. 

We conducted our search using PubMed as the only database. However, we retrieved a high number of records, and due to the repetition of the key contents in the included articles, we felt to have reached adequate content saturation. 

We applied abridged procedures for the screening process of the articles using the document search function. However, as the review intends to give a rather broad overview of the concept use and application rather than a comprehensive analysis of its contextual contents, we do not consider this approach as limiting for our results and conclusions. 

Regarding the qualitative analysis, we sought to elicit how authors aimed to define the concepts of our interest. However, we did not judge whether their definition seemed appropriate or “correct”.

Another limitation of our review is that we have not conducted an in-depth investigation of the barriers and enablers of the implementation of these three concepts. While this has been conducted in other fields, further research on the barriers and enablers specific to the public health workforce might be of additional value for overcoming challenges in the practical implementation of these concepts. As the majority of the papers describe the application of the three concepts in a research context, the conclusions we can draw for the public health workforce may lie primarily in the area of research and may be less applicable for a non-scientific context. 

We would also like to stress that all the review authors work primarily in the context of the public health workforce in Europe and thus have their main expertise in this region. Despite these limitations, our review is, to our best knowledge, the first review focussing on the definitions and applications of the concepts of multi-, inter-, and transdisciplinarity with respect to the public health workforce. A better understanding of how these concepts are applied by and to the public health workforce is crucial when aiming to tackle major current and future public health challenges by enhancing multiple disciplinary work. 

## 5. Conclusions

Definitions of multi-, inter-, and transdisciplinarity in the academic literature about the public health workforce were rare and heterogeneous, indicating uncertainty among authors and some conceptual overlap. Multiple disciplinary work can help address complex public health challenges and the three concepts should not be interpreted as mutually exclusive but as complementary, each with its own merit, costs, implementation challenges, and implications for collaboration. Hence, the careful selection of the appropriate mode of collaboration is advisable and the respective multiple disciplinary work mode should be made explicit to facilitate the collaboration. More integrative collaboration, such as in transdisciplinary work, may prove attractive for staff and can thus serve as a means to addressing public health workforce shortages. 

## Figures and Tables

**Figure 1 ijerph-19-10902-f001:**
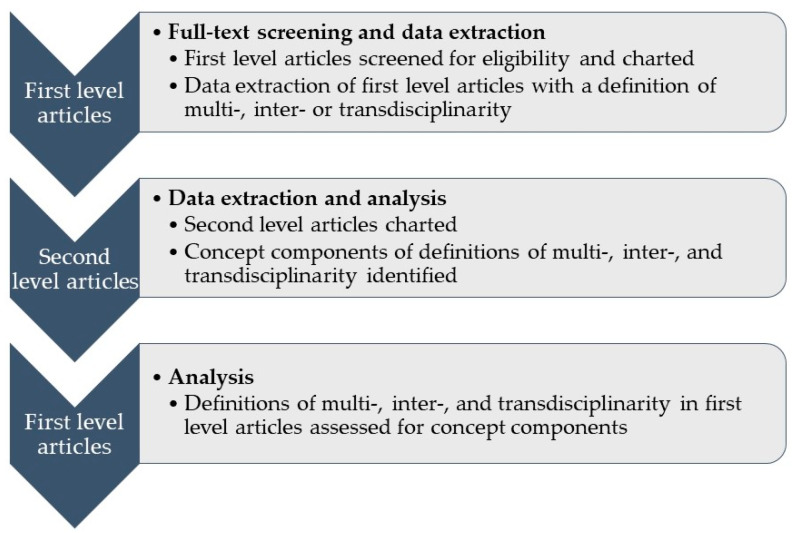
Overview of the data charting and analysis process.

**Figure 2 ijerph-19-10902-f002:**
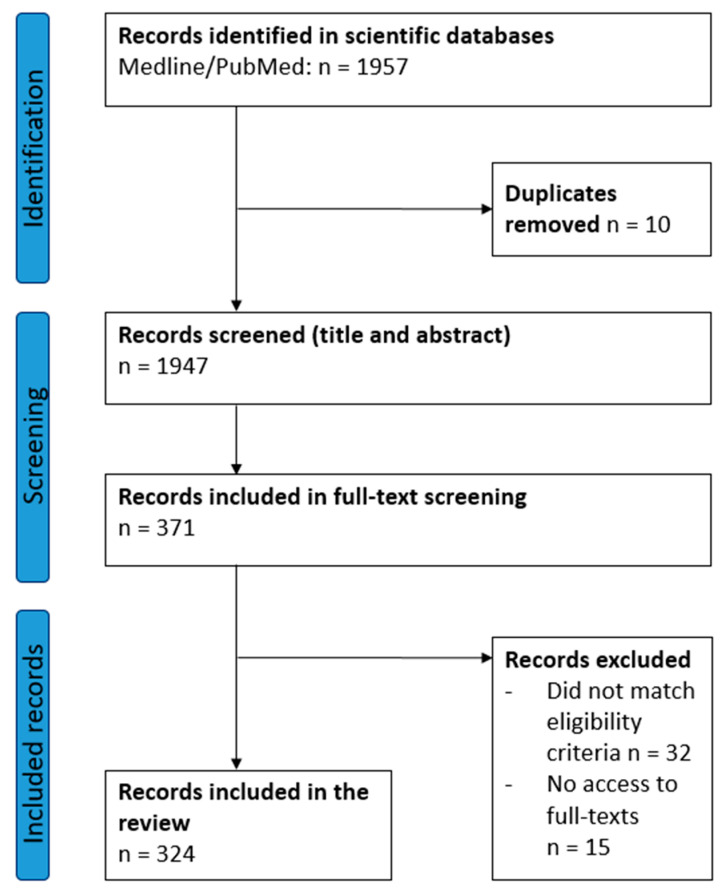
PRISMA flowchart.

**Figure 3 ijerph-19-10902-f003:**
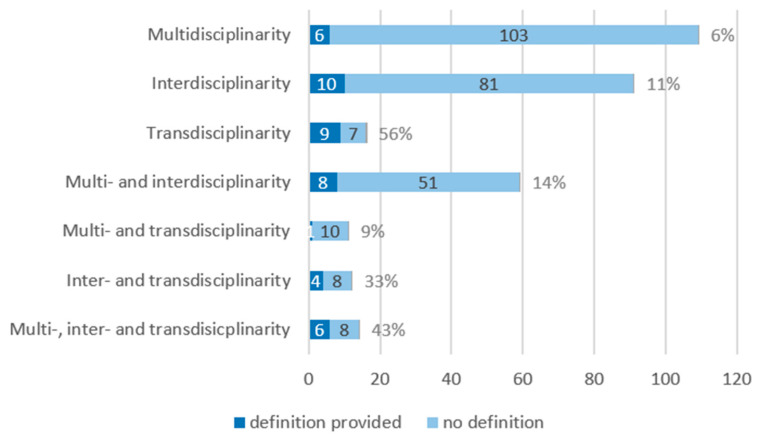
Concepts mentioned and defined in included articles (*n* = 324).

**Table 1 ijerph-19-10902-t001:** Descriptive statistics of concept mentions across included articles (*n* = 324).

Concept	Multidisciplinarity	Interdisciplinarity	Transdisciplinarity
Total mentions	193	176	53
Single mentions *	109	91	16
Combined mentions:			
*Multidisciplinarity*	NA	59	11
*Interdisciplinarity*	59	NA	12
*Transdisciplinarity*	11	12	NA

* Single mentions: only one of the concepts was mentioned in the respective article; fourteen articles mentioned all three concepts (not included in table).

**Table 3 ijerph-19-10902-t003:** Concept components of definitions of multi-, inter-, and transdisciplinarity in the theoretical literature and descriptions.

Concept Component	Description
Involved disciplines and/or professions	Concept component describing the number and type of individuals or groups of individuals involved in multiple disciplinary work.
Mode of collaboration	Concept component describing the processes, working mode, and methods of multiple disciplinary work.
Aim and purpose	Concept component describing the aims and objectives as well as general purpose and/or (potential) outcomes of multiple disciplinary work.
Role of participants	Concept component describing the role participants from different disciplines have in multiple disciplinary work.
Disciplinary boundaries	Concept component describing the maintenance, blurring, or dissolution of disciplinary boundaries in multiple disciplinary work.
Other	Other particularities described within definitions of multiple disciplinary work, e.g., requirements and challenges.

## Data Availability

Not applicable.

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
