# Peer review of "Multi-, Inter-, and Transdisciplinarity within the Public Health Workforce: A Scoping Review to Assess Definitions and Applications of Concepts"

_ijerph, 2022, doi:10.3390/ijerph191710902_

Round 1

Reviewer 1 Report

Thank you for letting me review your paper.

It is a very interesting topic and it is evident a lot of work has going into the search.

I do, however, have some questions or observations for you.

1.       I have never seen the word ‘actor’ used before in the context of working together professionally. Is this a commonly accepted term?

2.       In some places you use actor, and then you also talk about disciplines, professions etc. (e.g. line 378-380) If there is a possibility to use one consistent term this would make it easier for the reader. Using so many terms makes it complicated to follow for the reader.

3.       Or if you are going to use all terms can you better define them, or perhaps link to specific terms e.g. multi, inter or trans?

4.       Line 47. Full stop needed after et al

5.       In your introduction you say the public health workforce is not directly impacting on patients. You then also say that multi-professional work is patient facing roles working together. Therefore, can you justify including ‘multi’ in your search? Your results also showed this was less prominent. Your rationale also only focuses on inter and trans disciplinary work. Please justify using the search term multi in your rationale

6.       Line 115 – which content analysis method did you use?

7.       Why did you choose to only use one scientific database?

8.       Section 2.7 I believe would be better placed in the discussion as it is a reflection, and not a method that was employed.

9.       In the results, for some areas e.g. lines 313-314 you have put the article numbers, but you haven’t until that point. I would like to see more references added when papers are referred to throughout the results.

10.   In the results do countries of origin of the papers affect the definitions seen? Are they similar globally?

Were any of the definitions affected when translation occurred from German to English? How many papers needed to be translated for inclusion in the paper?

It is rare to have references in a conclusion as the conclusion would normally summarise your findings. You may wish to rewrite this focusing on the key concepts from this particular study.

General comment – please just take a second look to review American English versus UK English. You have some words in American English and others in UK English (s versus z). There are not many but it is best to be consistent, where possible.

Reviewer 2 Report

I consider that it is a well-executed review article and that it meets the characteristics established by PRISMA. However, a table is missing that clearly shows the authors and purpose of all the articles reviewed. Conclusions and discussion are clear and concise.

Round 2

Reviewer 1 Report

Thank you for answering all of my previous queries comprehensively.

I am happy that all my queries have been addressed. Good luck with the publication of this article.